# ERK3 Increases Snail Protein Stability by Inhibiting FBXO11-Mediated Snail Ubiquitination

**DOI:** 10.3390/cancers16010105

**Published:** 2023-12-24

**Authors:** Seon-Hee Kim, Ki-Jun Ryu, Keun-Seok Hong, Hyemin Kim, Hyeontak Han, Minju Kim, Taeyoung Kim, Dong Woo Ok, Jung Wook Yang, Cheol Hwangbo, Kwang Dong Kim, Jiyun Yoo

**Affiliations:** 1Division of Applied Life Science, Research Institute of Life Sciences, Gyeongsang National University, Jinju 52828, Republic of Korea; chenaliii@naver.com (S.-H.K.); ryu8650@naver.com (K.-J.R.); hongs06@naver.com (K.-S.H.); gpals8564@naver.com (H.K.); entreluzyluz@naver.com (H.H.); kimminju0091@naver.com (M.K.); ta0213@naver.com (T.K.); odw0205@naver.com (D.W.O.); chwangbo@gnu.ac.kr (C.H.); kdkim@gnu.ac.kr (K.D.K.); 2Department of Pathology, Gyeongsang National University Hospital, Gyeongsang National University College of Medicine, Jinju 52727, Republic of Korea; woogi1982@gnu.ac.kr; 3Division of Life Science, Gyeongsang National University, Jinju 52828, Republic of Korea

**Keywords:** ERK3, Snail, protein stability, FBXO11, pancreatic cancer

## Abstract

**Simple Summary:**

Several kinases are known to enhance the stability of the Snail protein, a key regulator of the epithelial-mesenchymal transition (EMT), by preventing its ubiquitination to induce the EMT process; however, the precise molecular mechanisms by which these kinases prevent Snail ubiquitination remain unclear. In this study, we found that the ERK3 kinase interacts with Snail and increases Snail protein stability by preventing ubiquitination in pancreatic cancer cells. Moreover, we found that ERK3 inhibits the binding of Snail to FBXO11, an E3 ubiquitin ligase that can induce Snail ubiquitination and degradation, resulting in increased Snail expression. Our findings suggest that the ERK3-Snail axis is a potential therapeutic target in pancreatic cancer.

**Abstract:**

Snail is a key regulator of the epithelial-mesenchymal transition (EMT), the key step in the tumorigenesis and metastasis of tumors. Although induction of Snail transcription precedes the induction of EMT, the post-translational regulation of Snail is also important in determining Snail protein levels, stability, and its ability to induce EMT. Several kinases are known to enhance the stability of the Snail protein by preventing its ubiquitination; however, the precise molecular mechanisms by which these kinases prevent Snail ubiquitination remain unclear. Here, we identified ERK3 as a novel kinase that interacts with Snail and enhances its protein stability. Although ERK3 could not directly phosphorylate Snail, Erk3 increased Snail protein stability by inhibiting the binding of FBXO11, an E3 ubiquitin ligase that can induce Snail ubiquitination and degradation, to Snail. Importantly, functional studies and analysis of clinical samples indicated the crucial role of ERK3 in the regulation of Snail protein stability in pancreatic cancer. Therefore, we conclude that ERK3 is a key regulator for enhancing Snail protein stability in pancreatic cancer cells by inhibiting the interaction between Snail and FBXO11.

## 1. Introduction

Cancer-related deaths are typically attributable to locally invasive tumor growth and/or the distant metastasis of tumor cells [1]. During metastasis, the epithelial cells initially lose apical-basal polarity and cell-cell contact while transforming to a mesenchymal phenotype [2]. This loss of epithelial features is often accompanied by increased cell motility and expression of mesenchymal genes, a process collectively referred to as the epithelial-mesenchymal transition (EMT). Indeed, EMT is a key step in the progression of tumors to the stage of metastasis [3,4]. Thus, research on the regulators of EMT is a contemporary hotspot in cancer research. One such regulator is the zinc finger protein Snail, which induces EMT by directly suppressing E-cadherin transcription during tumorigenesis or tumor progression [5,6].

Snail expression at the transcriptional level is controlled by various growth factors and cytokines such as HGF, TNF-α, and TGF-β [7,8,9]; however, even in the absence of the activation of these signaling pathways, Snail mRNA is consistently present in numerous cell types [10]. Snail is a highly unstable protein, and sub-cellular level or protein stability is governed by diverse kinases. GSK3β, for instance, hinders Snail function through phosphorylation, leading to nuclear export and ubiquitination-dependent cytosolic degradation [10,11,12]. Conversely, certain kinases boost Snail function by promoting nuclear import, nuclear retention, and enhancing its stability [13,14,15,16,17,18].

To date, four kinases (ATM, DNA-PKcs, Erk2, and p38) are known to enhance the stability of Snail, preventing its ubiquitination [15,16,17,18]; however, the molecular mechanisms by which these kinases prevent Snail ubiquitination are not well characterized. Here, we demonstrate that ERK3 inhibits the binding of FBXO11, an E3 ubiquitin ligase that induces Snail ubiquitination and degradation [19], resulting in the increased stability of Snail protein.

## 2. Materials and Methods

### 2.1. Cell Culture

HEK293T and all of the pancreatic cancer cell lines (AsPC-1, BxPC-3, CFPAC-1, MIA PaCa-2, and PANC-1) used in this study were obtained from the Korean Cell Line Bank (KCLB, Seoul, South Korea), where they were characterized by DNA-fingerprinting and isozyme detection. HEK293T, CFPAC-1, MIA PaCa-2, and PANC-1 cells were cultured in DMEM (Sigma-Aldrich, St. Louis, MO, USA) supplemented with 10% FBS and 1% penicillin and streptomycin. AsPC-1 and BxPC-3 cells were cultured in RPMI (Sigma-Aldrich, St. Louis, MO, USA) supplemented with 10% FBS and 1% penicillin and streptomycin.

### 2.2. Plasmid Construction and Transfection

The complete human Snail gene was inserted into a pDONR207 vector through the Gateway cloning system (Invitrogen, Carlsbad, CA, USA), adhering to the manufacturer’s guidelines. Subsequently, the entry clones were transformed into various destination vectors, namely pDEST-Flag-C and pDEST-HA-C. Flag-ERK3 (pCMV3-MAPK6-Flag), HA-ERK3 (pCMV3-MAPK6-HA), Myc-Snail (pCMV3-Snail-Myc), Myc-FBXO11 (pCMV3-FBXO11-Myc), Myc-FBXL5 (pCMV3-FBXL5-Myc), and Myc-FBXL14 (pCMV3-FBXL14-Myc) were procured from Sino Biological (Beijing, China). For transient transfection, HEK293T, MIA PaCa-2, and PANC-1 cells were cultured in 6-well or 100 mm diameter dishes for 24 h and transfected with the specified plasmid using X-tremeGENE 9 or X-tremeGENE HP transfection reagent, following the manufacturer’s instructions. After 48 h, the cells were collected and utilized for Western blot analysis. siRNA oligo duplexes targeting ERK3 (siERK3-1; 5′-CAUGAUUGGCCUGUACAUA-3′ and siERK3-2; 5′-GCUGUCCACGUACUUAAUUUA-3′) and FBXO11 (siFBXO11; 5′-UAGUCCAUACCAACUUCGUAGAAAA-3′) were obtained from Bioneer (Daejeon, South Korea). The transient transfection of siRNA oligo duplexes was carried out using Lipofectamine RNAiMAX Reagent (Invitrogen, Carlsbad, CA, USA), as per the manufacturer’s instructions.

### 2.3. Antibodies and Reagents

The mouse anti-Flag antibody was purchased from Applied Biological Materials (Richmond, BC, Canada). Rabbit anti-HA and rabbit anti-Snail antibodies were purchased from Cell Signaling Technology (Beverly, MA, USA). The mouse anti-α-tubulin antibody was purchased from Sigma-Aldrich (St. Louis, MO, USA). The rabbit anti-ERK3 antibody was purchased from Abcam (Cambridge, UK). The mouse anti-Myc antibody was purchased from Proteintec (Rosemont, IL, USA). Horseradish peroxidase (HRP)-conjugated anti-HA and HRP-conjugated anti-Flag antibodies were purchased from Sigma-Aldrich (St. Louis, MO, USA). HRP-conjugated anti-Myc was purchased from Millipore (Burlington, VT, USA). The proteasome inhibitor MG132 and translation inhibitor cycloheximide were purchased from Calbiochem (SanDiego, CA, USA).

### 2.4. Point Mutagenesis

Site-directed point mutagenesis was performed with an EZchange™ Site-directed Mutagenesis kit (Enzynomic, Daejeon, South Korea). The mutant strand synthesis reaction was performed by PCR. The reaction mixture contained a 10× reaction buffer, template plasmid, forward primer, reverse primer, dNTP mix, nPfu-Forte DNA polymerase, and sterile water. The PCR conditions were as follows: 94 °C for 5 min (initial denaturation), 94 °C for 30 s (denaturation), 55 °C for 1 min (annealing), 72 °C for 5 min (elongation) for 30 cycles, and 72 °C for 10 min (final elongation). Template removal and the PCR product ligation reaction were carried out according to the manufacturer’s instructions.

### 2.5. Western Blot Analysis

The cells were harvested after transfection and lysed in lysis buffer (20 Mm Tris pH 7.4, 2 mM EDTA, 150 mM sodium chloride, 1 mM sodium deoxycholate, 1% Triton X-100, 10% glycerol, 2 pills protease inhibitor cocktail (Roche, Basel, Switzerland)). The samples were mixed by vortexing and incubated for 30 min on ice, then heat-denatured proteins were loaded in equal volumes and separated on an 8–11% SDS-PAGE gel. After SDS-PAGE loading, the samples were transferred onto a PVDF membrane (Millipore, Burlington, VT, USA), and blocked with 5% non-fat dry milk. The membranes were incubated with the indicated primary antibodies overnight at 4 °C. After washing with TBS-T (0.1% Tween-20 contained TBS), the PVDF membranes were incubated with corresponding horseradish peroxidase (HRP)-conjugated secondary antibodies (1:2500) for 1 h at room temperature. The blots were developed with an enhanced chemiluminescence (ECL; BIO-RAD, Hercules, CA, USA) reaction, according to the manufacturer’s instructions. The primary antibodies used for Western blot analysis were as follows: anti-Flag antibody (1:1000 dilution, Applied Biological Materials, Richmond, BC, Canada), anti-HA antibody (1:1000 dilution, Cell Signaling Technology, Beverly, MA, USA), anti-Snail antibody (1:1000 dilution, Cell Signaling Technology, Beverly, MA, USA), anti-FBXO11 antibody (1:1000 dilution, Cell Signaling Technology, Beverly, MA, USA), anti-α-tubulin antibody (1:20,000 dilution, Sigma-Aldrich, St. Louis, MO, USA), anti-ERK3 antibody (1:1000 dilution, Abcam, Cambridge, UK), anti-HA antibody (1:5000 dilution, Sigma-Aldrich, St. Louis, MO, USA), anti-Flag antibody (1:1000 dilution, Sigma-Aldrich, St. Louis, MO, USA) and anti-Myc antibody (1:5000 dilution, Millipore, Burlington, VT, USA).

### 2.6. Immunoprecipitation

The cell lysates were incubated with the indicated antibodies overnight at 4 °C with gentle mixing, and then precipitated with Protein A+G agarose beads (Santa Cruz Biotechnology, Dallas, TX, USA) for 3 h at 4 °C. The beads were then washed five times with lysis buffer and eluted with 2× SDS sample buffer. Western blot analysis was then performed. The primary antibodies used for immunoprecipitation were as follows: anti-Flag antibody (Applied Biological Materials, Richmond, BC, Canada; 5 μg/mL), anti-HA antibody (Cell Signaling Technology, Beverly, MA, USA; 5 μg/mL), anti-Snail antibody (Cell Signaling Technology, Beverly, MA, USA; 5 μg/mL), and anti-Myc antibody (Proteintech; 5 μg/mL).

### 2.7. Total RNA Extraction and RT-PCR

Total RNA was extracted from the cultured cells using an RNeasy Mini Kit (Qiagen, Valencia, CA, USA). The cultured cells were collected into a 1.5 µL micro tube and washed twice with 1× PBS. The cells were lysed in 350 µL of RLT buffer containing 100 mM 2-merchaptoethanol (Sigma-Aldrich, St. Louis, MO, USA) and homogenized by passing the lysates at least five times through a 20-gauge needle. 350 µL of 70% ethanol was added to the homogenized lysates and mixed well by pipetting. The mixture was transferred into an RNeasy mini column and centrifuged at 12,000 rpm for 30 sec. After centrifugation, the RNeasy column was washed with RW1 buffer and RPE buffer by centrifugation at 12,000 rpm for 10 min. The total RNA on column was dissolved in 0.1% diethylpyrocarbonate (DEPC)-treated distilled water. The amount of RNA was measured by spectrometric absorbance at 260 nm. For the determination of further exact RNA concentrations, an equal amount of RNA was loaded on a 1.2% RNA gel in 1× TAE buffer. RT-PCR was executed employing an AccuPower^®^ RT-PCR PreMix kit from Bioneer. In each AccuPower^®^ RT-PCR PreMix tube, 250 ng of total RNA and specific primers were combined, and RNase-free water was supplemented to achieve a total volume of 20 µL. The Thermo Electron PCR thermal cycler was employed for the RT-PCR process. The resulting amplified products were segregated on 1.5% agarose gels. The RT-PCR protocol included the following steps: 45 °C for 30 min (reverse transcription), 94 °C for 5 min (inactivation of RTase), 94 °C for 1 min, 52 °C for 1 min, 72 °C for 1 min for 25 cycles, succeeded by a 5 min incubation at 72 °C. The primers used were as follows: Snail (F: 5′-ATGACTGAAAAAGCCCCA-3′; R: 5′-TCATTCTGTCCACTCCTT-3′), β-actin (F: 5′-GTGGGGCGCCCCAGGCACCA-3′; R: 5′-CTCCTTAATGTCACGCACGAT-3′).

### 2.8. Immunofluorescence Analysis

The cells were seeded on a confocal dish (SPL Life Sciences, Pocheon, South Korea) at a density of 2.5 × 10^5^ cells. 48 h after transfection, the cells were rinsed three times with PBS and fixed for 20 min at room temperature in 4% paraformaldehyde. The fixed cells were permeabilized with PBS containing 0.1% Triton X-100 for 20 min, washed three times in cold PBS, and blocked with blocking solution (1% BSA in PBS) for 1 h at room temperature. After blocking, the cells were incubated with the primary anti-Flag antibody (mouse, 1:1000 dilution, Applied Biological Materials, Richmond, BC, Canada) and anti-Snail antibody (rabbit, 1:1600 dilution, Cell Signaling Technology, Beverly, MA, USA) in 1% BSA overnight at 4 °C. Following three washes with cold PBS, the cells were incubated with the anti-mouse IgG-TRITC antibody (1:200 dilution, Sigma-Aldrich, St. Louis, MO, USA) and anti-rabbit IgG-FITC antibody (1:200 dilution, Sigma-Aldrich, St. Louis, MO, USA) for 1 h at room temperature. The labeled cells were rinsed three times with cold PBS, mounted in VECTASHIELD^®^ Antifade Mounting Medium with DAPI (Vector Laboratories, Burlingame, CA, USA). The cells were examined by a multiphoton and laser scanning confocal microscope (FV1000MPE, Olympus, Shinjuku, Tokyo, Japan).

### 2.9. Cycloheximide Chase Assay

The cells were seeded on a 6-well plate at a density of 2 × 10^5^ cells per well. After culturing overnight, the cells were transfected with plasmids as desired. 48 h after transfection, 50 μg/mL of cycloheximide (CHX) was treated. The cells were collected at 0 min, 30 min, 60 min, 90 min, and 120 min following treatment with cycloheximide and were subjected to Western blot analysis.

### 2.10. Ubiquitination Assay

The ubiquitination assay was done following an immunoprecipitation protocol. HEK293T cells were transfected with plasmids expressing HA-Ub, HA-Ub-K48, or HA-Ub-K63 with Myc-Snail and Flag-ERK3 WT or mutants (ERK3 CA and ERK3 KD). Two days after transfection, the cells were treated with a 5 μM proteasome inhibitor MG132 (Calbiochem, San Diego, CA, USA) for 12 h to inhibit the proteasomal degradation of the Snail protein before lysed with a Triton X-100 lysis buffer. The cell lysates were split for use as input and 1.5 mg of cell lysates were collected for immunoprecipitation. The cell lysates for immunoprecipitation were incubated with the anti-Myc antibody (Proteintec) overnight at 4 °C with gentle mixing, and then precipitated with Protein A+G agarose beads (Santa Cruz Biotechnology, Dallas, TX, USA) for 3 h at 4 °C. The beads were then washed five times with lysis buffer and eluted with 2× SDS sample buffer. Followed by Western blot analysis with the anti-HA antibody (1:5000 dilution, Sigma-Aldrich, St. Louis, MO, USA), anti-Myc antibody (1:5000 dilution, Millipore, Burlington, VT, USA) and anti-Flag antibody (1:1000 dilution, Sigma-Aldrich, St. Louis, MO, USA), and detection was carried out with an enhanced chemiluminescence (ECL; BIO-RAD, Hercules, CA, USA) reaction, according to the manufacturer’s instructions.

### 2.11. In Vitro Kinase Assay

The recombinant GST-ERK3 was purchased from SignalChem (Richmond, BC, Canada). As substrates, MBP-Snail-WT (full length), MBP-Snail-S11A (full length), or MBP-Snail S246A (full length) were purified from E. coli cultures with MBP Excellose^®^ beads (Bioprogen, Daejeon, South Korea). GST-ERK3 and MBP-Snail WT, MBP-Snail S11A, or MBP-Snail S246A were incubated in kinase buffer (25 mM Tris-HCl pH7.5, 5 mM β-glycerophosphate, 2 mM dithiothreitol (DTT), 0.1 mM Na_3_VO_4_, 10 mM MgCl_2_) and ^32^P-labeled ATP (BMS, Seoul, South Korea) for 30 min at 30 °C. The samples were boiled for 5 min and analyzed by SDS-PAGE.

### 2.12. GST Pull-Down Assay

Recombinant HIS-FBXO11 and HIS-ERK3 were purchased from Proteintech (Rosemont, IL, USA). GST or GST-Snail was purified from E. coli cultures with Glutathione HiCap Matrix beads (Qiagen, Valencia, CA, USA). Recombinant HIS-FBXO11 was incubated with either GST or GST-Snail in the presence or absence of HIS-ERK3 overnight at 4 °C in GST binding buffer (50 mM Tris-HCl, pH 7.5, 150 mM NaCl, 1 mM EDTA). The complex, captured using Glutathione HiCap Matrix beads, was analyzed by SDS-PAGE.

### 2.13. Pancreatic Cancer Tissue Specimens

De-identified human pancreatic cancer tissue specimens (37 cases), preserved in paraffin, were gathered between 2001 and 2012 at Gyeongsang National University Hospital in Jinju, South Korea. Pathologists at Gyeongsang National University Hospital examined and diagnosed these clinical pancreatic cancer tissue specimens. The collection and analysis of tumor samples received approval (Approved Number: 2018-04-010) from the Institutional Review Board (IRB) at Gyeongsang National University Hospital with a waiver of informed consent.

### 2.14. Immunohistochemistry

From individual formalin-fixed paraffin-embedded tissues, 3 mm diameter cores were taken and then placed in fresh paraffin blocks from the recipient. Analysis focused on two tissue cores from the most representative tumor areas. Immunohistochemistry (IHC) was conducted on 4 μm thick paraffin sections utilizing a BenchMark ULTRA from Ventana Medical Systems Inc. (Oro Valley, AZ, USA) and the Optiview DAB IHC Detection Kit (Ventana Medical Systems Inc., Oro Valley, AZ, USA). Polyclonal antibodies specific to Snail (1:750) and monoclonal antibodies for ERK3 (Cambridge, UK, 1:50) were applied for IHC. ULTRA Cell Conditioning 1 (Ventana Medical Systems Inc., Oro Valley, AZ, USA) served for antigen retrieval for Snail and ERK3, with a primary antibody incubation time of 32 min.

### 2.15. Statistical Analysis

Statistical analysis utilized PASW Statistics 18.0 software from IBM Corporation (Somers, NY, USA). Data are presented as means ± SD. Significance of differences was assessed through the chi-square test. Pairwise comparisons were conducted using Student’s *t*-test. In cases of multiple comparisons, data underwent testing via one-way ANOVA, followed by the Dunnett post-hoc test. *p* values of less than 0.05 were deemed statistically significant.

## 3. Results

### 3.1. ERK3 Interacts with Snail in Pancreatic Cancer Cells

In our previous study, we identified several kinases that can influence Snail function by using yeast two-hybrid screening with a home-made human kinase cDNA library [18]. In this study, we focused on ERK3, which is known to increase the migratory and invasive abilities of various types of cancer cells [20,21]. We first confirmed the interaction between ERK3 and Snail by reciprocal immunoprecipitation in HEK293T cells co-transfected with vectors encoding HA-Snail and/or Flag-ERK3 (Figure 1A). Next, we confirmed the interaction between endogenous ERK3 and Snail in MIAPaCa-2 and PANC-1 pancreatic cancer cells (Figure 1B). To identify the Snail motif that is required for the interaction with ERK3, we co-expressed five truncated forms of Snail along with full-length ERK3 and subjected the resulting cells to IP using an anti-Flag antibody. In this experiment, all ERK3 mutants except the zinc finger domains deletion mutant (ΔZnF 1-4) retained the ability to associate with ERK3 (Figure 1C). These findings suggested that ERK3 interacts with the zinc finger domains of Snail. We next investigated the subcellular localization of ERK3 and Snail and found that, when ERK3 was co-expressed with Snail, these proteins were co-localized mainly in the nucleus (Figure 1D).

### 3.2. Correlation of ERK3 Expression Levels with Clinical-Pathological Features of Pancreatic Cancer

To investigate the clinical significance of ERK3 expression levels in various cancer patients, we first analyzed GEPIA (Gene Expression Profiling Interactive Analysis) data and found increased expression levels of ERK3 in the majority of cancer types (Figure 2A). A GEPIA survival heat map indicated that a higher expression of the ERK3 gene was associated with poor survival in the context of several cancers (Figure 2B). These analyses indicated the potential prognostic value of ERK3 expression in pancreatic cancer. Therefore, we further investigated the effect of ERK3 expression in patients with pancreatic cancer. Analysis of three different public microarray data sets (GSE62165 from GEO; Pei and Segra from Oncomine) revealed a higher expression of ERK3 in pancreatic cancer tissues compared to normal tissues (Figure 2C). Analysis of The Cancer Genome Atlas (TCGA) datasets indicated a progressive increase in ERK3 expression levels with the advancing stage of pancreatic cancer (Figure 2D), and that pancreatic cancer patients with poor prognosis have high ERK3 expression (Figure 2E). Indeed, the survival rate of pancreatic cancer patients showed a negative correlation with the ERK3 expression levels. Higher ERK3 levels were associated with lower overall survival and relapse-free survival rates (Figure 2F).

### 3.3. Correlation of ERK3 and Snail Protein Levels in Pancreatic Cancer Patients and Cell Lines

To verify the correlation between ERK3 and Snail protein expression levels in clinical samples, we conducted a tissue microarray analysis of 37 pancreatic cancer tissue specimens and observed a strong positive correlation between ERK3 and Snail expression (Figure 3A). Tumors lacking ERK3 expression exhibited minimal or no Snail expression. In contrast, tumors with strong ERK3 expression exhibited strong nuclear Snail expression. Analysis of public microarray data set (GSE26088 from GEO) indicated an inverse correlation between the expression levels of ERK3 and E-cadherin (one of the negatively regulated Snail target genes) in pancreatic xenografts and cell lines (Figure 3B). These results suggested that an elevated expression of ERK3 may repress E-cadherin expression levels in patients with pancreatic cancer by enhancing Snail protein levels.

### 3.4. ERK3 Increases Snail Protein Stability by Suppressing Ubiquitination-Dependent Snail Degradation in Pancreatic Cancer Cells

Since several kinases are known to regulate Snail protein stability [15,16,17,18], we next investigated whether ERK3 affects the stability of Snail protein. We noticed a significant increase in HA-Snail expression in HEK293T cells after co-transfection with Flag-ERK3 (Figure 1A, left). Before determining the effect of ERK3 expression on Snail protein stability, we confirmed the relative protein expression levels of ERK3 and Snail in various pancreatic cancer cell lines (Appendix A). In addition, since Snail is a highly unstable protein that is rapidly degraded by the proteasome [10], Snail protein levels were investigated in pancreatic cancer cell lines after treatment with the proteasome inhibitor MG132. As expected, MG132 treatment increased the levels of Snail protein in most cell lines but, in Panc-1 cells, which had the highest basal Snail protein levels, MG132 treatment barely increased Snail protein levels (Appendix A). Therefore, in this study, we used MiaPaCa-2 cells to upregulate Snail and Panc-1 cells to downregulate Snail.

The expression of ERK3 significantly increased the endogenous Snail protein levels in HEK293T and MiaPaCa-2 cells (Figure 4A), while having no effect on mRNA-expression levels (Appendix A). The RNAi-mediated depletion of ERK3 expression decreased the endogenous Snail protein level in Panc-1 cells (Figure 4B) without affecting Snail mRNA levels (Appendix A). A cycloheximide (CHX) pulse-chase analysis revealed that ERK3 overexpression significantly increased the half-life of the Snail protein in HEK293T and MiaPaCa-2 cells (Figure 4C). In contrast, the depletion of ERK3 expression significantly decreased the half-life of Snail protein in Panc-1 cells (Figure 4D). Collectively, these findings indicated that ERK3 may increase the stability of Snail protein in pancreatic cancer cells.

ERK3 also increased the level of exogenously expressed Snail in HEK293T cells (Figure 4E). Notably, inhibiting proteasome function with MG132 significantly increased the exogenous Snail expression to the same level irrespective of ERK3 expression (Figure 4E). These results suggested that ERK3 may increase Snail protein stability by inhibiting proteasome-dependent Snail degradation. Moreover, ERK3 expression was also found to significantly decrease Snail ubiquitination (Figure 4F). Overall, these findings suggested that ERK3 may increase Snail protein stability by suppressing ubiquitination-dependent Snail degradation.

To understand the role of ERK3 in Snail function, we next examined whether ERK3 might contribute to Snail-promoted EMT. For this purpose, we made Flag-tagged ERK3 transfectants from MIA PaCa-2 cells, and found that the morphology of ERK3-expressing cells was distinct from that of control cells. Confocal microscopy analysis of phalloidin-stained cells showed that MIA PaCa-2 cells overexpressing ERK3 transformed into mesenchymal cells (Figure 4G). These findings prompted us to examine the expression levels of EMT marker genes in these cells. Concordantly, ERK3 expression inhibited the expression of E-cadherin, one of the representative epithelial marker proteins, while promoting the expression of N-cadherin, one of the representative mesenchymal marker proteins, in MIA PaCa-2 cells (Figure 4H). These findings indicated that ERK3 overexpression can induce EMT in pancreatic cancer cells.

### 3.5. ERK3 Kinase Activity Is Not Essential for Increasing the Stability of Snail Protein

Considering the observed interaction between ERK3 and Snail, we hypothesized that Snail could be a direct substrate of the ERK3 kinase. A search for the consensus ERK3 phosphorylation motif (RXXSXXS) unveiled two potential Erk3-phosphorylation motifs in Snail, located at Ser 11 and 246 (Appendix A). To determine whether ERK3 can phosphorylate Snail, we performed an in vitro kinase assay using commercially available recombinant active ERK3 kinase and purified maltose binding protein (MBP)-fused wild-type (WT) or mutant Snail proteins. Active ERK3 was not found to phosphorylate WT and mutant Snail proteins in vitro, but myelin basic protein, one of the well-known substrates for pan-MAPK, was phosphorylated by ERK3 kinase (Appendix A). These results suggested that Snail is not a direct substrate for ERK3 kinase.

Since we already confirmed that ERK3 interacts with Snail and regulates its protein stability, we investigated whether ERK3 kinase activity affects this regulation. For this purpose, we used two different mutant ERK3 kinase constructs: constitutive active (CA)-ERK3 (Ser189 residue of ERK3 that phosphorylated and activated by PAK1 was replaced by phosphor-mimetic aspartic acid) and kinase dead (KD)-ERK3 (Asp171 residue of ERK3 in ATP binding pocket was replaced by alanine) [22]. We found that these two mutants ERK3 (CA and KD) could interact with Snail (Figure 5A) and enhance exogenously expressed Snail protein levels in HEK293T cells (Figure 5B). In addition, the expression of ERK3 mutants (CA and KD) markedly enhanced the endogenous Snail protein levels in HEK293T and MiaPaCa-2 cells, similar to wild-type (WT)-ERK3, without affecting their mRNA expression levels (Figure 5C). The cycloheximide (CHX) pulse-chase analysis demonstrated that mutants ERK3 (CA and KD), as well as WT, also markedly extended the half-life of the endogenous Snail protein in HEK293T cells (Figure 5D). Furthermore, the expression of these mutants (CA and KD) significantly decreased Snail protein ubiquitination, similar to WT-ERK3 (Figure 5E). Collectively, these results suggested that ERK3 can increase the stability of Snail protein regardless of ERK3 kinase activity.

### 3.6. ERK3 Increases Snail Expression Levels by Inhibiting FBXO11 Binding to Snail

PAK1 is known to phosphorylate Snail at Ser246 residue and contribute to its translocation from the cytoplasm to the nucleus [13]. To determine the mechanism by which ERK3 increases Snail protein stability, independent of kinase activity, we first examined the possibility that ERK3 serves as a scaffolding protein for PAK1 to phosphorylate Snail. To this end, we generated a Snail mutant construct (S246A) that could not be phosphorylated by PAK1 and observed that ERK3 increased S246A-Snail expression to the same level as wild-type Snail (Figure 6A). These results indicated that PAK1 is not relevant to the increased expression of Snail protein by ERK3.

Since ERK3 decreased Snail ubiquitination, we next examined whether ERK3 could promote the deubiquitination of Snail protein. For this purpose, we used the deubiquitinase inhibitors PR-619 and WP1130 and observed that treatment with these inhibitors did not suppress the ERK3-mediated increase in Snail protein levels (Figure 6B). These results indicated that ERK3 increases the expression of Snail protein regardless of the action of deubiquitinases.

Many ubiquitin ligases can regulate subcellular levels of Snail by increasing ubiquitination-dependent Snail degradation [10,19,23,24,25,26]. Since ERK3 reduced the ubiquitination of Snail protein without phosphorylation, we further examined whether ERK3 inhibits the binding of ubiquitin ligase(s) that can ubiquitinate Snail protein. Because ERK3 binds to the zinc finger domains of Snail protein, we selected ubiquitin ligases FBXL5, FBXL14, and FBXO11, which can ubiquitinate Snail by binding to the same domains (25,26,19), and found that only FBXO11 can reduce the expression of Snail protein (Figure 6C). However, FBXO11 did not suppress the ERK3-induced increase in Snail protein expression (Figure 6D). Furthermore, the interaction between FBXO11 and Snail was significantly inhibited by ERK3 in HEK293T cells (Figure 6E) and in vitro (Figure 6F). Finally, using ubiquitin mutants that can mediate only K48-linked or K63-linked ubiquitination, respectively, we found that FBXO11 increases K48-linked ubiquitination of Snail protein and that ERK3 inhibits it (Figure 6G,H). These results suggested that ERK3 increases Snail expression by inhibiting the binding of FBXO11 to Snail and the ubiquitination of Snail protein by FBXO11, which is important for Snail degradation.

### 3.7. FBXO11 Plays an Important Role in ERK3-Induced Increase in Snail Protein Stability in Pancreatic Cancer Cells

To determine whether ERK3 also increases Snail protein stability by inhibiting the binding of FBXO11 to Snail in pancreatic cancer cells, we used siRNA to suppress the expression of FBXO11 in MIA PaCa-2 cells, and then expressed ERK3. In MIA PaCa-2 cells, suppressing FBXO11 expression increased Snail expression, and expressing ERK3 after suppressing FBXO11 increased Snail expression, but not more than when FBXO11 expression was solely suppressed (Figure 7A). The stability of Snail protein was also increased when FBXO11 expression was inhibited, and inhibiting FBXO11 expression and expressing ERK3 increased the stability of Snail protein to a similar extent as inhibiting FBXO11 expression (Figure 7B). These results suggested that FBXO11 plays an important role in the ERK3-induced increase in Snail protein stability in pancreatic cancer cells.

## 4. Discussion

In our previous study, we screened several kinases that can influence Snail function by using yeast two-hybrid screening with a home-made human kinase cDNA library [18]. Among them, in this study, we identified the molecular mechanism by which ERK3 increases the stability of Snail protein. In normal or early-stage tumor cells with low ERK3 expression, FBXO11 binds to the zinc finger domains of Snail and degrades it via the ubiquitination-proteasome system (UPS) (Figure 8, left panel). In malignant tumor cells with high expression of ERK3, ERK3 increases the stability of Snail protein by inhibiting the binding between FBXO11 and Snail, and the stabilized Snail might induce the EMT process (Figure 8, right panel).

Similar to other signaling proteins, the activities of transcription factors are modulated by kinase(s) in response to diverse cellular signals. In the case of Snail, multiple kinases exert control over its protein stability and function through various mechanisms. The stability of the Snail protein undergoes negative regulation through ubiquitination-dependent proteasomal degradation in the cytoplasm. Consequently, facilitating its translocation into the nucleus can enhance its stability. P21-activated kinase 1 (PAK1) plays a role in this process by phosphorylating Snail at Ser246 within the zinc finger domain, a crucial step for its nuclear localization [27]. This phosphorylation event increases Snail’s protein expression and function [13]. Although the precise protein machinery involved in the nuclear translocation of Snail remains unclear, it is suggested that Snail phosphorylation by PAK1 likely enhances its interaction with proteins associated with its nuclear trafficking. We observed the co-localization of ERK3 with Snail in the nucleus (Figure 1D), and the intracellular localization of Snail protein remained unaltered with ERK3 overexpression; therefore, we concluded that ERK3 does not increase Snail protein stability by inducing its nuclear translocation from the cytoplasm.

Another way to increase Snail stability is by increasing Snail deubiquitination. Many deubiquitinases, including DUB3, OTUB1, USP1, USP3, USP11, USP18, USP26, USP37, and USP27X, remove ubiquitination from Snail proteins, thereby increasing their intracellular stability [28,29,30,31,32,33,34,35,36,37]. Some deubiquitinases can be phosphorylated by kinases to regulate their activity [38,39,40]. The phosphorylation of proteins either increases or decreases their binding force to the substrates. For instance, in the case of USP4, phosphorylation by Akt increases its interaction with a transforming growth factor-β (TGF-β) type I receptor [38]. Conversely, the phosphorylation of USP25 by vaccinia-related kinase 2 (VRK2) decreases its interaction with TRiC [39], and phosphorylation of USP21 by Erk1 decreases its interaction with Nanog [40]. However, we found that ERK3 increased Snail expression in the presence of DUB inhibitors (Figure 6B); therefore, we also ruled out the possibility that ERK3 increases Snail protein stability through the action of DUB.

A third potential way is by preventing Snail ubiquitination. Phosphorylation at Ser100 by ATM and DNA-PKcs decrease Snail ubiquitination or its interaction with GSK3β, both of which are critical for its degradation, thereby increasing its stability [15,16]. Erk2-mediated Snail phosphorylation at Ser82/Ser104 residues was shown to protect it from ubiquitination and subsequent proteasomal degradation [17]. We previously proposed that p38 MAPK increases Snail protein stability by suppressing the dual specificity tyrosine-phosphorylation-regulated kinase 2 (DYRK2)-mediated prime phosphorylation of GSK3β, which is critical for βTrCP-mediated Snail ubiquitination and subsequent degradation [18]. However, Erk3 was not found to phosphorylate Snail (Appendix A) and ERK3 kinase activity was not essential for increasing Snail protein stability (Figure 5); therefore, we hypothesized that ERK3 would inhibit the binding of ubiquitin ligase(s) to Snail, which is involved in lowering the stability of Snail protein, and found that ERK3 increased Snail protein stability by inhibiting FBXO11 binding to Snail.

## 5. Conclusions

In conclusion, our study reveals a potential molecular mechanism by which ERK3 may regulate the metastatic ability of several cancer cells by increasing Snail protein stability. Our study not only unravels a critical mechanism underlying ERK3-mediated Snail protein stability, but also has important implications for the development of treatment strategies for metastatic cancers.

## Figures and Tables

**Figure 1 cancers-16-00105-f001:**
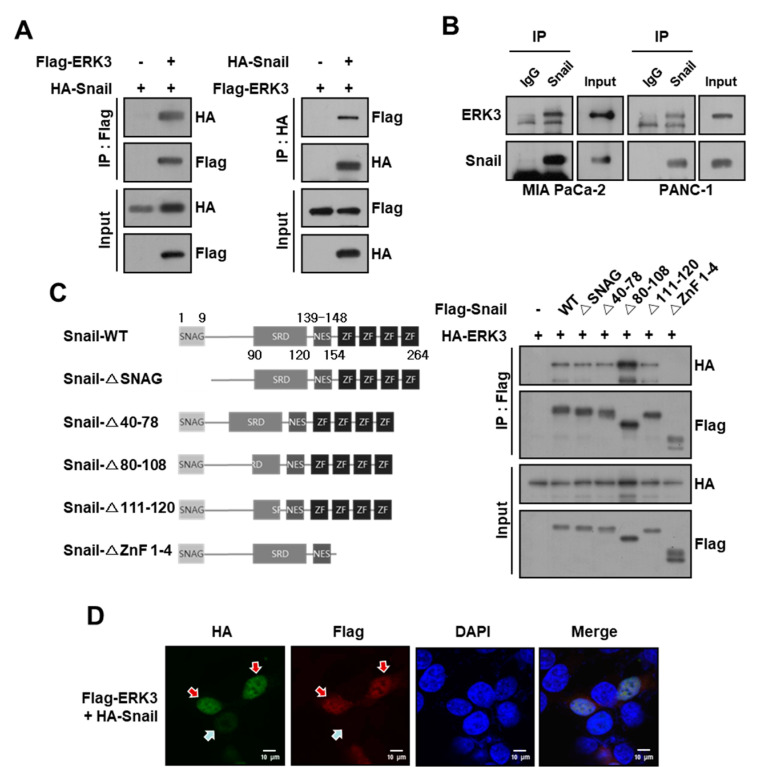
ERK3 interacts with Snail in pancreatic cancer cells. (**A**) The co-immunoprecipitation assay in HEK293T cells co-transfected with HA-Snail and Flag-ERK3 plasmids. (**B**) The immunoprecipitation assay in MIA PaCa-2 and PANC-1 pancreatic cancer cells with the anti-Snail antibody. (**C**) Left, schematic diagram for Snail deletion mutants. Right, interaction between ERK3 and Snail deletion mutants. HEK293T cells co-transfected with HA-ERK3 and the Flag-tagged deletion mutants of Snail plasmids. Immunoprecipitation with the anti-Flag antibody, followed by Western blot with the anti-HA antibody. The uncropped blots are shown in Appendix A. (**D**) The subcellular localization of ERK3 and Snail. HEK293T cells were transfected with HA-Snail or co-transfected with Flag-ERK3 and HA-Snail. The transfected cells were fixed and examined by confocal microscopy. Superimposing the two colors (merge) indicates the co-localization of ERK3 and Snail. Red arrows indicate Snail protein levels in the presence of ERK3; the blue arrow indicates Snail protein levels in the absence of ERK3.

**Figure 2 cancers-16-00105-f002:**
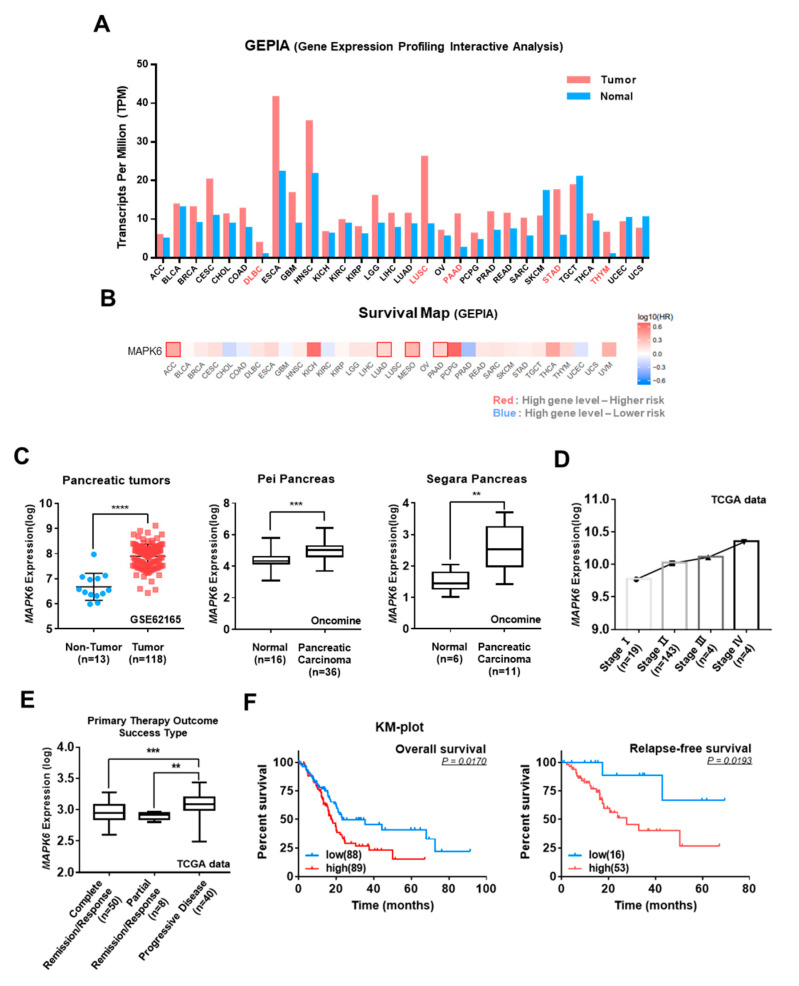
The correlation of ERK3 expression levels with the clinical-pathological features of pancreatic cancer. (**A**) The ERK3 gene expression levels in various normal and tumor samples obtained from GEPIA (Gene Expression Profiling Interactive Analysis). The height of the bar represents the median expression of a certain tumor type or normal tissue. The tumor types indicated in red are significance. (**B**) The survival heat map obtained from GEPIA. A color closer to red means a higher ERK3 gene expression level leads to higher risk and blue means a higher ERK3 gene expression level leads to lower risk. A red square line indicates significance. (**C**) ERK3 mRNA levels in pancreatic tumors. Left, GSE62165 data series obtained from the GEO datasets of NCBI (National Center for Biotechnology Information). Middle and right, the Pei and Segara datasets of Oncomine. Statistical significances were determined by *t*-test. (**D**) The ERK3 mRNA expression of each pancreatic cancer stage. (**E**) ERK3 mRNA expression according to primary therapy outcome success type. The database from TCGA (The Cancer Genome Atlas). **, *p* < 0.01, ***, *p* < 0.001, ****, *p* < 0.0001 as determined by *t*-test. (**F**) Overall survival rate and relapse-free survival rate of pancreatic cancer patients. The survival database was analyzed from KM-plot (Kaplan–Meier plotter).

**Figure 3 cancers-16-00105-f003:**
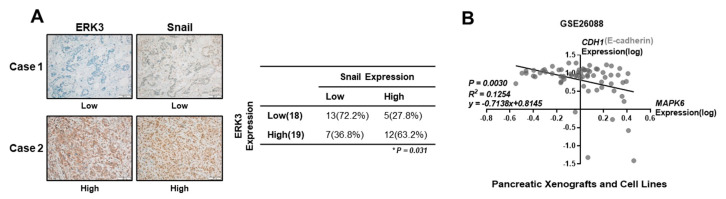
The correlation of ERK3 and Snail protein levels in pancreatic cancer patients and cell lines (**A**) Left, representative IHC images of ERK3 and Snail in pancreatic tumors. Scale bars, 100 μm. Right, correlation of the Snail expression level with ERK3 levels in pancreatic cancer patients. Statistical significances were determined by Chi-squared test. (**B**) The correlation of ERK3 mRNA expression and E-cadherin mRNA expression in pancreatic cancer patient-derived xenograft (PDX) mouse models and human pancreatic cancer cell lines. GSE26088 data series obtained from the GEO datasets of NCBI. Statistical significances were determined by linear regression analysis.

**Figure 4 cancers-16-00105-f004:**
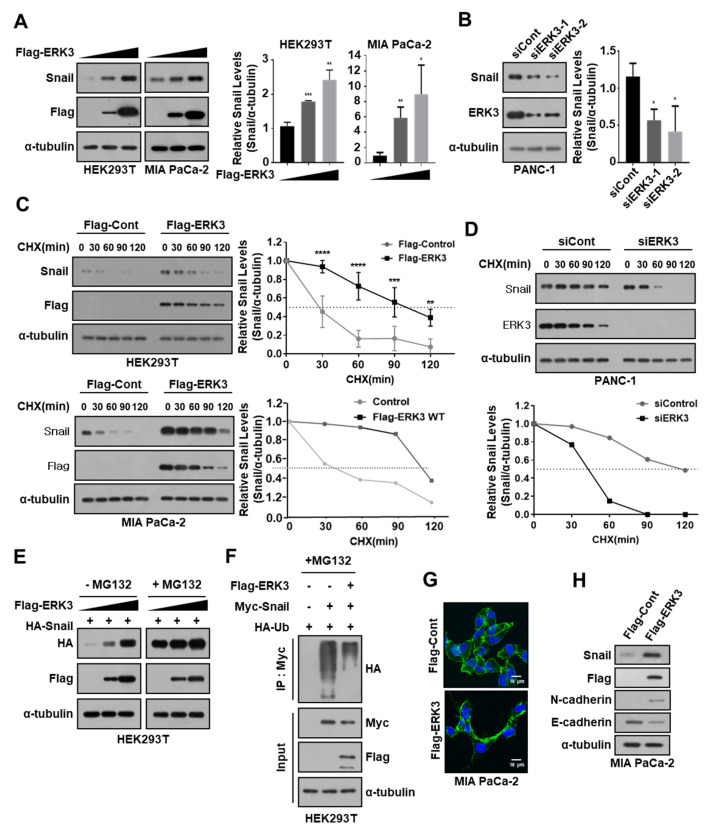
ERK3 increases Snail protein stability by suppressing ubiquitination-dependent Snail degradation in pancreatic cancer cells. (**A**) Dose-dependent up-regulation of endogenous Snail protein levels by ERK3. HEK293T and MIA PaCa-2 cells were transfected with increasing concentrations of Flag-ERK3. Western blot was performed with anti-Snail and anti-Flag antibodies (left) and quantified Snail protein levels from Western blot images (right). (**B**) Snail protein levels decreased by ERK3 depletion. PANC-1 cells were transfected with two different siRNA specific for ERK3, respectively. The expression levels of endogenous Snail and ERK3 protein were determined by Western blot with anti-Snail and anti-ERK3 antibodies (left) and quantified Snail protein levels from Western blot images (right). (**C**) The increase of endogenous Snail protein stability by ERK3. HEK293T and MIA PaCa-2 cells were transfected with plasmids expressing Flag-ERK3 and treated with 20 μg/mL cycloheximide for the indicated times before harvest, and Western blot was performed with Snail- and Flag-specific antibodies. The right panel presents the mean ± SD of the densitometric analyses of Snail levels in three independent experiments. * *p* < 0.05, **, *p* < 0.01, ***, *p* < 0.001, ****, *p* < 0.0001 as determined by *t*-test. (**D**) Destabilization of endogenous Snail by ERK3 depletion. The PANC-1 cells transfected with ERK3-specific siRNA were treated with 20 μg/mL cycloheximide for the indicated times before harvest, and Western blot was performed with Snail- and ERK3-specific antibodies. Snail protein levels were quantified from Western blot images. (**E**) HA-Snail was co-transfected with increasing concentrations of Flag-ERK3 and treated with MG132 (10 μM) for 12 h before harvest. Western blot was performed with anti-HA and anti-Flag antibodies. (**F**) Myc-Snail was co-transfected with HA-Ub (HA-Ubiquitin) and Flag-ERK3 into HEK293T cells, followed by treatment with 5 μM MG132 for 12 h. Cell lysates underwent immunoprecipitation using an anti-Myc antibody and were subsequently analyzed through immunoblotting with a specific antibody against HA. The presented data are representative of three independent experiments, yielding consistent results. (**G**) The morphological changes of ERK3-expressing MIA PaCa-2 cells. The ERK3-overexpressing MIA PaCa-2 and control cells visualized by confocal microscopy after staining with FITC-conjugated phalloidin. (**H**) The expression levels of EMT marker proteins in ERK3-expressing MIA PaCa-2 and control cells were analyzed by immunoblotting. The uncropped blots are shown in Appendix A.

**Figure 5 cancers-16-00105-f005:**
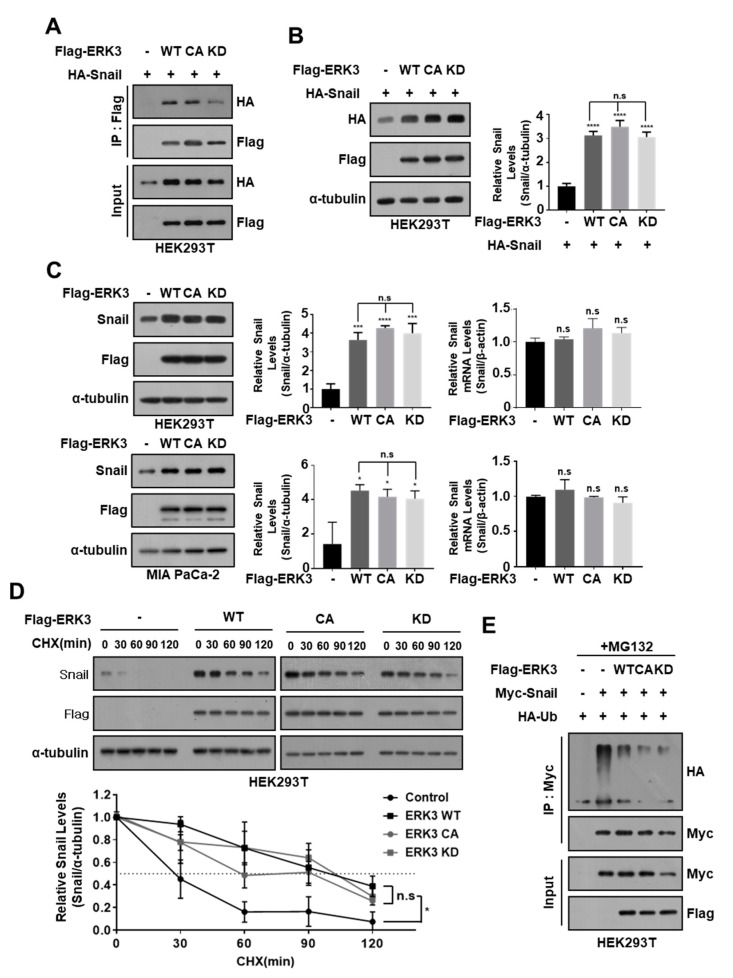
ERK3 kinase activity is not essential for increasing the stability of Snail protein. (**A**) HA-Snail was co-transfected with plasmids expressing Flag-ERK3 WT, CA, or KD into HEK293T cells. The cell lysates were immunoprecipitated using an anti-Flag antibody and then analyzed by immunoblotting using a specific antibody against HA. The data are representative of three independent experiments with similar results. (**B**) HA-Snail was co-transfected with plasmids expressing Flag-ERK3 WT, CA, or KD into HEK293T cells. The expression levels of exogenous Snail and ERK3 protein were determined by Western blot with anti-HA and anti-Flag antibodies (left) and quantified Snail protein levels from Western blot images (right). n.s.: no significance, *, *p* < 0.05, *** *p* < 0.001, ****, *p* < 0.0001 as determined by *t*-test. (**C**) The HEK293T and MIA PaCa-2 cells were transfected with plasmids expressing Flag-ERK3 WT, CA, or KD. The expression levels of endogenous Snail were determined by Western blot with anti-Snail antibodies (left) and quantified Snail protein levels from Western blot images (middle). The mRNA level of Snail was analyzed by qRT-PCR (right). (**D**) The increase of endogenous Snail protein stability by Flag-ERK3 WT, CA, or KD. HEK293T cells were transfected with plasmids expressing Flag-ERK3 WT, CA, or KD, and treated with 20 μg/mL cycloheximide for the indicated times before harvest, and Western blot was performed with Snail- and Flag-specific antibodies. The right panel presents the mean ± SD of the densitometric analyses of Snail levels in three independent experiments. *, *p* < 0.05 as determined by *t*-test. (**E**) Myc-Snail was co-transfected with HA-Ub (HA-Ubiquitin) and Flag-ERK3 WT, CA, or KD into HEK293T cells, and then treated with 5 μM MG132 for 12 h. The cell lysates were immunoprecipitated using an anti-Myc antibody and then analyzed by immunoblotting using a specific antibody against HA. The data are representative of three independent experiments with similar results. The uncropped blots are shown in Appendix A.

**Figure 6 cancers-16-00105-f006:**
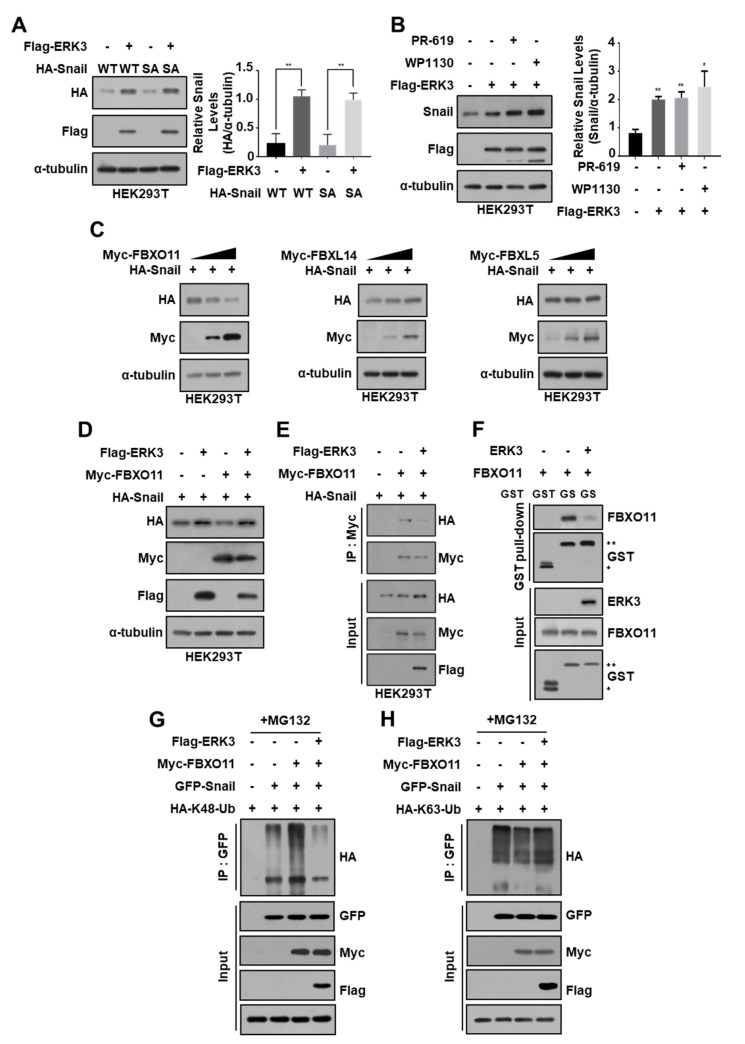
ERK3 increases Snail expression levels by inhibiting FBXO11 binding to Snail. (**A**) HA-Snail WT or HA-Snail S246A (SA) was co-transfected with plasmid expressing ERK3 into HEK293T cells. The expression levels of exogenous Snail and ERK3 protein were determined by Western blot with anti-HA and anti-Flag antibodies (left) and quantified Snail protein levels from Western blot images (right). **, *p* < 0.01 as determined by *t*-test. (**B**) ERK3 was transfected into HEK293T cells and then treated with 20 μM PR-619 or 10 μM WP1130 for 6 h. The expression levels of endogenous Snail protein were determined by Western blot with anti-Snail antibodies (left) and quantified Snail protein levels from Western blot images (right). *, *p* < 0.06, **, *p* < 0.01 as determined by *t*-test. (**C**) The HEK293T cells were transfected with HA-Snail and increasing concentrations of Myc-FBXO11, FBXL14, or FBXL5. Western blot was performed with anti-Snail and anti-Myc antibodies. (**D**) HA-Snail was co-transfected with plasmids expressing FBXO11 and/or ERK3 into HEK293T cells. The cell lysates were immunoblotted with the indicated antibodies. (**E**) The interaction between exogenous FBXO11 and Snail in the presence or absence of ERK3. HA-Snail and Myc-FBXO11 were co-transfected with or without Flag-ERK3 into HEK293T cells. The cell lysates were immunoprecipitated with an anti-Myc and analyzed by Western blot using the anti-HA antibody. (**F**) The in vitro interaction of Snail and FBXO11 in the presence or absence of ERK3. Recombinant FBXO11 was incubated with GST or GST-Snail in the presence or absence of recombinant ERK3. The complex, captured using Glutathione HiCap Matrix beads, was analyzed by Western blot using the anti-FBXO11 antibody. *, GST; **, GST-Snail. (**G**,**H**) GFP-Snail was co-transfected with HA-K48-Ub (**G**) or HA-K63-Ub (**H**), Myc-FBXO11, and Flag-ERK3 into HEK293T cells, followed by treatment with 5 μM MG132 for 12 h. Cell lysates underwent immunoprecipitation using an anti-GFP antibody and were subsequently analyzed through immunoblotting with a specific antibody against HA. The presented data are representative of three independent experiments, yielding consistent results. The uncropped blots are shown in Appendix A.

**Figure 7 cancers-16-00105-f007:**
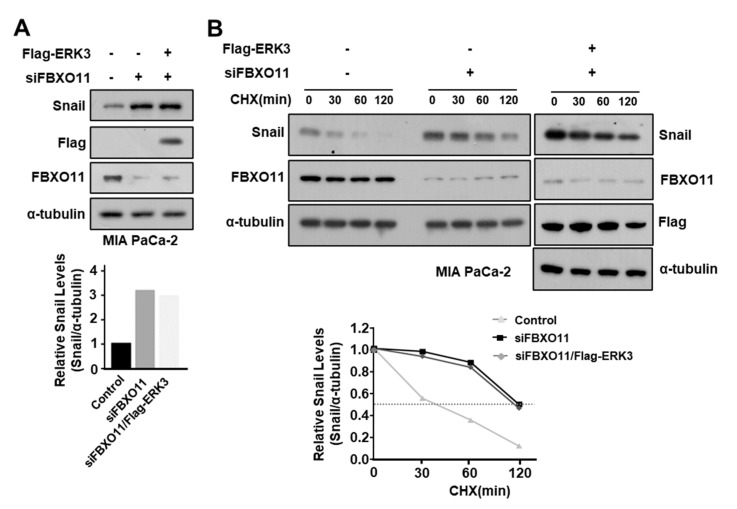
FBXO11 plays an important role in the ERK3-induced increase in Snail protein stability in pancreatic cancer cells. (**A**) The effect of the FBXO11 on ERK3-induced increase in Snail protein expression in MIA PaCa-2 cells. MIA PaCa-2 cells transfected with FBXO11-specific siRNA were transfected with Flag-ERK3. The expression levels of endogenous Snail protein were determined by Western blot with the anti-Snail antibody (upper) and quantified from Western blot images (lower). (**B**) The effect of FBXO11 on the ERK3-induced increase in Snail protein stability in MIA PaCa-2 cells. MIA PaCa-2 cells transfected with FBXO11-specific siRNA were transfected with Flag-ERK3. The cells were treated with 20 μg/mL cycloheximide for the indicated times before harvest, and Western blot was performed with the Snail-specific antibody (upper). Snail protein levels were quantified from Western blot images (lower). The uncropped blots are shown in Appendix A.

**Figure 8 cancers-16-00105-f008:**
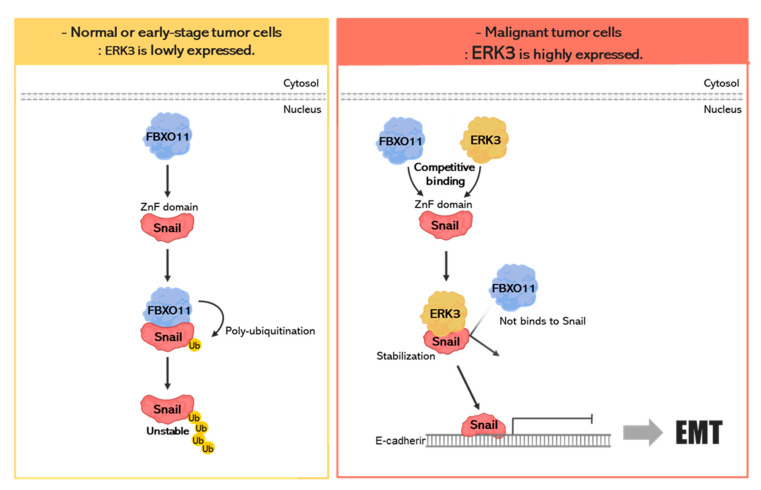
Proposed model to illustrate how the stabilization of Snail by ERK3 may lead to EMT.

## Data Availability

The data presented in this study are available on request from the corresponding authors.

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
