# Peer review of "ERK3 Increases Snail Protein Stability by Inhibiting FBXO11-Mediated Snail Ubiquitination"

_cancers, 2023, doi:10.3390/cancers16010105_

Round 1
Reviewer 1 Report
Comments and Suggestions for Authors
In this manuscript, Kim and colleagues reported that ERK3 stabilizes Snail, an EMT-inducing factor, by directly binding to Snail. Mechanistic study indicates that ERK3 stabilizes Snail by inhibiting FOXO11-mediated Snail ubiquitination, which is independent of ERK3 kinase activity. Overall, the discovery is novel and the manuscript is suitable for publication at Cancers if the following issues are properly addressed.
1. More evidence are needed to demonstrate that FOXO11 mediates the effects of ERK3 on Snail. Can ERK3 over-expression or knockdown regulates Snail protein levels and Snail ubiquitination in the absence of FOXO11?
2. Authors showed that ERK3 inhibits the binding of FOXO11 to Snail using a co-IP assay (Figure 6). Can authors observe similar phenotype in a pull-down assay by using recombinant proteins? Can ERK3 inhibits Snail ubiquitination by FOXO11 in an in vitro ubiquitination assay?
Author Response
Dear reviewer
Thank you very much for taking the time to review this manuscript. Please find the detailed responses below and the corresponding revisions/corrections highlighted/in track changes in the re-submitted files.
In this manuscript, Kim and colleagues reported that ERK3 stabilizes Snail, an EMT-inducing factor, by directly binding to Snail. Mechanistic study indicates that ERK3 stabilizes Snail by inhibiting FOXO11-mediated Snail ubiquitination, which is independent of ERK3 kinase activity. Overall, the discovery is novel and the manuscript is suitable for publication at Cancers if the following issues are properly addressed.
Point 1: More evidence are needed to demonstrate that FOXO11 mediates the effects of ERK3 on Snail. Can ERK3 over-expression or knockdown regulates Snail protein levels and Snail ubiquitination in the absence of FOXO11?
Response 1: I appreciate the reviewer's perspective, and I would like to respectfully address any potential misunderstandings. Our study has provided evidence demonstrating that the overexpression of ERK3 in the absence of FOXO11 indeed leads to an increase in both the expression and stability of Snail, as depicted in Figure 4A and 4C. Additionally, our findings indicate that the inhibition of ERK3 expression results in a decrease in both the expression and stability of Snail, as illustrated in Figure 4B and 4D. We also confirmed that overexpression of ERK3 in Mia PaCa-2 cells induced EMT and added the results to Figure 4G and 4H.
Point 2: Authors showed that ERK3 inhibits the binding of FOXO11 to Snail using a co-IP assay (Figure 6). Can authors observe similar phenotype in a pull-down assay by using recombinant proteins? Can ERK3 inhibits Snail ubiquitination by FOXO11 in an in vitro ubiquitination assay?
Response 2: We sincerely appreciate the constructive advice provided by the reviewer and fully acknowledge its merit, as highlighted earlier. Regrettably, we are unable to carry out the suggested experiments due to limitations in our available materials. Nevertheless, in response to the reviewer's valuable input, we have conducted alternative experiments. Specifically, we have discovered that FOXO11 promotes Lys-48-linked polyubiquitination of Snail proteins, and ERK3 functions as an inhibitor of this process. These additional results have been thoughtfully incorporated into Figure 6F and 6G. We believe that these findings contribute valuable insights to the understanding of the regulatory mechanisms involving FOXO11, ERK3, and Snail.
Reviewer 2 Report
Comments and Suggestions for Authors
“In the present manuscript, the authors demonstrated that ERK3 is a key regulator for enhancing Snail protein stability in pancreatic cancer cells by inhibiting the interaction between Snail and FOXO11. Authors presented generally acceptable data. Put together, most data were suitably supporting author’s hypothesis. However, there are several concerns about your data and careful considerations is needed.”
1. The author should detect changes in EMT related proteins to confirm changes in cellular morphology.
2. The author should test the effect of ERK3 on ubiquitination of snail protein K48 or K63 sites.
3. In Figure 4C, MIAPaCa-2 cells were transfected with plasmids expressing Flag-ERK3 and treated with 20μg/ml cycloheximide, why does Snail protein expression increase at 30 min and 60 min after CHX treatment.
Author Response
Dear reviewer
Thank you very much for taking the time to review this manuscript. Please find the detailed responses below and the corresponding revisions/corrections highlighted/in track changes in the re-submitted files.
In the present manuscript, the authors demonstrated that ERK3 is a key regulator for enhancing Snail protein stability in pancreatic cancer cells by inhibiting the interaction between Snail and FOXO11. Authors presented generally acceptable data. Put together, most data were suitably supporting author’s hypothesis. However, there are several concerns about your data and careful considerations is needed.
Point 1: The author should detect changes in EMT related proteins to confirm changes in cellular morphology.
Response 1: We express our sincere gratitude for the thoughtful advice offered by the reviewer. Following the reviewer's guidance, we established a cell line that overexpresses ERK3 in MIA PaCa-2 cells. Our findings indicate that these cells demonstrate an enhanced epithelial-mesenchymal transition (EMT) in comparison to control cells. This is supported by notable alterations in cell morphology and the expression of EMT marker proteins. The pertinent results have been thoughtfully incorporated into Figures 4G and 4H to further strengthen and illustrate our study. We hope that these additional findings contribute positively to the overall quality and comprehensiveness of our manuscript.
Point 2: The author should test the effect of ERK3 on ubiquitination of snail protein K48 or K63 sites.
Response 2: We acknowledge our agreement with the points mentioned earlier. In response to the reviewer's suggestions, we have conducted additional experiments revealing that FOXO11 plays a crucial role in promoting Lys-48-linked polyubiquitination of Snail proteins. Furthermore, we have identified ERK3 as a key inhibitor of this process. To ensure a comprehensive presentation of our findings, these supplementary results have been integrated into Figures 6F and 6G. We believe that these enhancements significantly contribute to the depth and clarity of our study.
Point 3: In Figure 4C, MIAPaCa-2 cells were transfected with plasmids expressing Flag-ERK3 and treated with 20μg/ml cycloheximide, why does Snail protein expression increase at 30 min and 60 min after CHX treatment.
Response 3: We hold the view that the observed increase in Snail protein expression in the initial experiment may be attributed to an experimental artifact. To address this concern, we have repeated the experiment, and the revised results have been substituted in the lower section of Figure 4C.
Round 2
Reviewer 1 Report
Comments and Suggestions for Authors
Authors did not address my concerns at all.
Point 1. More evidence are needed to support that FOXO11 mediates the effects of ERK3 on Snail: cell lines used in Figure 4 should contain endogenous FOXO11. If FOXO11 is knocked down or knocked out, can authors still observe similar phenotype shown in Figures 4A-D?
Point 2. To demonstrate a competition binding model, the best experiment design is using recombinant proteins to show the competition binding among ERK3, FOXO11 and Snail in vitro, and to show that ERK3 can inhibit Snail ubiquitination by FOXO11 in vitro.
Author Response
Dear reviewer
Thank you very much for taking the time to review this manuscript. Please find the detailed responses below and the corresponding revisions/corrections highlighted/in track changes in the re-submitted files.
Point 1. More evidence are needed to support that FOXO11 mediates the effects of ERK3 on Snail: cell lines used in Figure 4 should contain endogenous FOXO11. If FOXO11 is knocked down or knocked out, can authors still observe similar phenotype shown in Figures 4A-D?
Response 1: I appreciate the reviewer's perspective and would like to express my regret for any misunderstandings. Following the reviewer's suggestions, we conducted experiments, which revealed that the expression and stability of the Snail protein were enhanced by Erk3, even when suppressing the expression of FOXO11 using siRNA in MIA PaCa-2 cells. These findings indicate a significant role for FBXO11 in the ERK3-induced increase in Snail protein stability in pancreatic cancer cells. These additional results have been thoughtfully incorporated into new Figure 7. We believe that these findings contribute valuable insights to the understanding of the regulatory mechanisms involving FBXO11, ERK3, and Snail.
Point 2. To demonstrate a competition binding model, the best experiment design is using recombinant proteins to show the competition binding among ERK3, FOXO11 and Snail in vitro, and to show that ERK3 can inhibit Snail ubiquitination by FOXO11 in vitro.
Response 2: We express our sincere gratitude for the thoughtful advice offered by the reviewer. Following the reviewer's suggestions, we conducted GST pull-down experiments, which revealed that the interaction between FBXO11 and Snail was inhibited by ERK3 in vitro. These results further solidify our experimental findings that ERK3 inhibits the binding of FBXO11 to Snail, thereby increasing the stability of the Snail protein. These additional results have been thoughtfully incorporated into Figure 6F.
Unfortunately, we were unable to perform another experiment suggested by the reviewer, the in vitro ubiquitination experiment, as it requires a lot of materials, which are not available at this time. Nevertheless, in response to the reviewer's valuable comments, we have conducted alternative experiments. Specifically, we have discovered that FBXO11 promotes Lys-48-linked polyubiquitination of Snail proteins, and ERK3 inhibits this process. These additional results have been thoughtfully incorporated into Figure 6G and 6H. We believe that these findings contribute valuable insights to the understanding of the regulatory mechanisms involving FBXO11, ERK3, and Snail.
Reviewer 2 Report
Comments and Suggestions for Authors
The authors have addressed my concerns and it is ready for publication.
Author Response
We express our sincere gratitude for the thoughtful advice offered by the reviewer.
Round 3
Reviewer 1 Report
Comments and Suggestions for Authors
Authors addressed all my concerns.